# Betaine: A Potential Nutritional Metabolite in the Poultry Industry

**DOI:** 10.3390/ani12192624

**Published:** 2022-09-30

**Authors:** Wafaa A. Abd El-Ghany, Daryoush Babazadeh

**Affiliations:** 1Poultry Diseases Department, Faculty of Veterinary Medicine, Cairo University, Giza 12211, Egypt; 2Faculty of Veterinary Medicine, Shiraz University, Shiraz P.O. Box 71964-84334, Iran

**Keywords:** antioxidant, betaine, heat stress, metabolism, osmolyte, performance, poultry

## Abstract

**Simple Summary:**

The effects of antibiotics as alternative feed additives on the health of poultry have gained promising results. Betaine has been extensively used in the poultry industry as a potential metabolite. This review article focuses on the interaction between betaine and other elements, such as methionine, choline, and creatine. In addition, poultry carcass traits and performance, as well as its osmoregulatory, anticoccidial, immune-modulatory, and heat-stress alleviation activities, were discussed with regard to the effects of dietary supplementation of betaine. Due to the different advantages of using betaine, it has been suggested to be a promising feed additive to the poultry diet. More research is needed to explain how betaine exerts its action in the body.

**Abstract:**

Poultry industry has been recognized as a fast-developing sector aiming to produce low-cost and high-nutrient foods for human consumption. This review article aimed to focus on the significant roles of dietary betaine supplementation in the poultry industry. In this respect, different effects of betaine on performance and carcass traits, as well as its osmoregulatory, anticoccidial, immune-modulatory, and heat-stress alleviation activities, were discussed. Different concentrations of betaine supplementation can improve the feed conversion ratio, final body weight, breast muscle yield, egg production ratio, and reduce body fat contents in broiler chicken, turkey, duck, geese, and quail diets. Betaine supplemented with methyl groups can eliminate the need to have some methyl-group donors, including choline and methionine, therefore having positive effects on feed conversion ratio in poultry diets. The osmolytic character of betaine can alleviate heat stress and have a positive impact on tonic immobility, which consequently reduces stress in poultry. By inhibiting distinct developmental stages of *Eimeria* species, betaine reduces the damaging effects of coccidiosis on broiler chickens and improves intestinal structure and function. The immunological, cardiovascular, neurological, renal, and hepatic metabolic systems benefit from betaine’s osmo-protective properties. Therefore, betaine has the potential to be considered as an alternative to feed additives and enhances the health status and productive performance of poultry.

## 1. Introduction

The poultry industry is an important, fast-growing sector for producing low-cost and rich nutrients for human consumption. This industry is always at risk of infectious and non-infectious agents that cause adverse losses. Antibiotics, especially growth promoter types, are widely used to increase gut health, reduce subclinical infections, and promote growth and production [1]. However, antibiotic overuse negatively impacts the balance of the normal gut microbiome along with the accumulation of antimicrobial residues in tissue and the presence of new strains of drug-resistant pathogenic bacteria [2]. Therefore, many potential feed alternatives have been investigated to substitute antibiotics as growth and health enhancers, immune stimulants, and antimicrobials [3,4]. Among different substitutes, phytogenic feed additives have been widely used to boost immunity and relieve stress [5].

Betaine is a general term used for trimethyl derivative of the amino acid glycine, a substance for betaine-homocysteine methyltransferase in the kidney and liver [6,7]. It is composed of three methyl groups with a hydrophobic nature and one carboxyl group with a hydrophilic nature. Betaine can be widely found in plants and animals with great physiological importance. It is a by-product of sugar beet processing, as can be found highly in sugar beets, the aqueous form of molasses. Although wheat and wheat bran contain considerable amounts of betaine, the substance is not largely observable in poultry feedstuffs, such as soybean and corn [8]. Both betaine and folic acid are methyl donors and can act as compensators for the lack of labile methyl groups in diets containing maize and soybean [9,10,11].

Studies have addressed betaine as a highly valuable phytobiotic dietary supplementation [12]. Betaine comes in a variety of forms, including anhydrous, monohydrate, and hydrochloride forms, all of which are beneficial to the poultry production system [13,14]. The required dietary concentration of betaine largely depends on the concentration of other labile methyl groups, environmental conditions, and the health condition of the birds. Betaine hydrochloride is approved by the European Food Safety Authority [15] and used as a trimethylamine product for broilers and layers in doses of 15 mg and 24 mg/kg feed, respectively. The efficacy of betaine decreases at higher concentrations [16]. One of the most recent studies indicated that high doses of anhydrous betaine or hydrochloride betaine were beneficial for broiler chickens [17]. Of note, metabolic responses of betaine may depend on time [18].

Dietary supplementation with betaine positively impacts growth performance, muscle yield, fat metabolism, and immunity [19,20]. Betaine has been used as a lipotropic agent to reduce and liberate fat from the liver and turnover lipids from the body [21]. It can also be used to handle broilers’ diets with low-crude protein (with high metabolized energy: crude protein ratio) [22]. Moreover, the osmo- regulatory effect of betaine is vital for the immune system, cardiovascular, nervous, and renal metabolic functions [13,23,24]. The osmolytic character of betaine can also help alleviate heat stress [25,26] and improve nutrient digestibility [27]. Betaine possesses osmo-regulatory properties because of its zwitterionic structure, which effectively protects intestinal enzymes and cell proteins from environmental stresses, and consequently, performance is improved [28]. Acute or chronic liver injuries, dysfunction, and failure could be mediated via either the direct effect of betaine on the mitochondria that alleviate the oxidative stress or via inhibition of satellite cell activities (Figure 1).

From the above mentioned, this review article aimed to focus on the significant roles of betaine as a nutritional supplement in the poultry industry. In this respect, the different effects of betaine on performance and carcass trait, as well as its osmoregulatory, anticoccidial, immune-modulatory, and heat-stress alleviation activities, were discussed.

## 2. Interaction of Betaine with Other Elements

### 2.1. Methionine

Betaine is used in trans-methylation processes to synthesize carnitine and creatine after the methyl group donation [13,14]. In addition, it can potentially eliminate the need for creatine, methionine, choline as methyl-group donors [29,30]. Betaine donates the methyl radicals for the remethylation of homocysteine to methionine and to formulate creatine, carnitine, and phosphatidylcholine through the S-adenosyl methionine pathway. A methyl group is transferred from betaine to dimethylglycine through the enzyme betaine homocysteine methyl transferase (BHMT) [31]. It also contains three methyl groups that allow effective spare of dietary methionine [32,33]. Homocysteine forms methionine either through betaine or 5-methyl tetrahydrofolate [31,34]. Being a methyl group donor, betaine may sometimes replace methionine in a reaction with homocysteine [35,36,37]. Thus, methionine could be used more for its growth function [38]. Dietary betaine can be used as an alternative to methionine for broiler chicks [39]. According to a study, supplemental methionine in the range of 30–80% could be substituted by betaine with no adverse effects on broilers’ performance [40]. However, some reports indicated non-significant results when methionine was partially or entirely substituted with betaine in the diet, which can be due to the dietary supply of cystine [41,42]. Dietary betaine and cystine mixture positively affected the feed conversion ratio (FCR) in broiler chicks, compared to supplementing diets with each of them separately [40,43].

### 2.2. Choline

Choline, a betaine precursor, is essential for some physiological purposes, including synthesizing a cell membrane and forming acetylcholine. Betaine is derived from choline in the body [44,45] during oxidization in the mitochondria of liver cells that later forms methionine [46]. As estimated in poultry, betaine may be used to supply 25% of the choline requirement [47]. In broiler chickens, the efficiency of choline in the methylation process of creatine is significantly less than betaine [48]. The use of dietary ionophore coccidiostats, which suppress choline oxidase activity, enhances the effectiveness of betaine in poultry [49]. Thus, a sufficient supply of labile methyl groups may require dietary supplementation of betaine.

### 2.3. Creatine

Creatine is active in the muscle energy buffering system, also known as “methyl guanidine acetic acid” is a nitrogen-containing chemical substance synthesized in the brain, kidney, pancreas, and liver, as a precursor for arginine, methionine, and glycine [50]. Creatine regulates cellular energy metabolism due to its role as a phosphate carrier and reservoir for adenosine triphosphate (ATP) synthesis. Betaine, as a methyl group donor, is very important in the transmethylation process for carnitine and creatine formation [13], and affects fat metabolism [51]. Moreover, it significantly affects lipid metabolism through synthesizing carnitine and creatine, known as methylated substances, in the muscles and livers of male broilers [52]. Betaine affects the muscles’ pH by increasing the muscle content with creatine [53]. It has been reported that increasing the level of carnitine [54] and creatine [55] in muscles [53] and creatinine in blood [56] depends on the capacity of betaine as a methyl donor. Dietary betaine and/or creatine have indicated significant efficacy in enhancing the water holding capacity, and consequently, efficient nutritional hyper-hydration process, thereby reducing the risk of broiler chicks exposed to high ambient temperature [57]. A recent study indicated that adding guanidino acetic acid and betaine to broiler ducks’ diets boosted their levels of creatine and ATP level in pectoral muscle as well as creatine kinase in the blood plasma, but reduced the activities of lactatedehydrogenase and L-arginine: glycine amidino transferase enzymes [58].

## 3. Different Beneficial Effects of Dietary Betaine in the Poultry Industry

### 3.1. Reproductive Performance

#### 3.1.1. Chickens

##### Broiler

Supplementation of poultry diet with betaine can enhance performance parameters, including feed intake, body weight, FCR, and body weight gain in broiler chickens [59,60,61,62,63] and ducks [64]. The impacts of dietary betaine supplementation on broiler chickens’ growth parameters are shown in Appendix A. Dietary betaine has revealed enhancement of performance as well as pectoral muscle yield in broiler chickens [28]. It can also boost the performance parameters in heat-stressed broiler chickens [20,26]. A significant increase in weight gain was found in broiler chickens exposed to heat stress and given dietary betaine with reduced methionine and choline chloride [65]. Likewise, supplementing broiler chickens’ diet with betaine increased their weight gain when exposed to high temperatures [66]. Compared to heat-stressed chickens, betaine-supplemented broiler chickens revealed higher feed intake and body weight gain, but lower FCR [67]. In another study, the results showed that broilers given betaine (2 g/kg diet) under heat stress had higher feed intake and body weight gain, compared to those fed on normal energy and low energy diets without betaine [68]. Broiler chickens exposed to disease conditions, such as coccidiosis, showed an enhancement of growth performance parameters [69] and feathers growth [70] after supplementation of betaine.

The positive impact of betaine on performance parameters might be attributed to its osmotic character that protects the intestinal epithelia, supports intestinal cell growth, enhances cell activity, improves intestinal morphology [71], and consequently increases nutrient digestibility [72]. Betaine enhances the absorption and utilization of nutrients by improving ether extract, crude fiber, crude protein, dry matter, and non-nitrogen fiber extract digestibility which can help the expansion of intestinal mucosa [73]. As a “methyl group donor”, betaine could improve weight gain and feed efficiency [61,63]. It conserves the energy required for the Na^+^/K^+^ pump at high temperatures and consumes this energy to fuel the growth of broiler turkeys [74]. Moreover, supplementing broilers’ diets with betaine improves sulfur amino acid availability [70].

Dietary betaine could also be regarded as an effective stimulus for the growth and multiplication of beneficial microbiota that protects the digestive tract [28,75]. Dietary supplementation of betaine could reduce the total crop bacterial count and increase the number of *Enterococci* in broilers [76]. In addition, betaine can increase the levels of total short-chain fatty acids (SCFAS) and blood electrolytes [77]. The SCFAS, including acetic and propionic acids, are essential for some beneficial intestinal bacteria, such as *Lactobacillus* and *Bifidobacterium*, in the gut of poultry. Accordingly, the inhibition of some intestinal colonization by harmful bacteria may occur [78]. Betaine at 4.5 g/kg of the diet does not affect the intestinal concentration of lactic acid. However, it significantly increases the intestinal SCFAS concentrations by 77.6 mmol/kg dry matter, suggesting intensified bacterial activity [6].

Broiler chickens that received dietary 0.05%, 0.05–0.10%, and 0.28% betaine showed no improvement in parameters of performance [10,79]. Dietary betaine and/or creatine supplementation during the rearing period of broiler chickens did not affect production parameters [57].

##### Layer

Adding betaine to the diet of layer chickens induced improvement in FCR [14] and egg production [80]. Under the stress of hot climates, betaine enhanced the productivity and laying parameters of laying hens. For instance, dietary supplementation of betaine at a concentration of 0.2% improved the heat-stressed chickens’ eggshell quality and egg production [81]. A recent study on laying hens confirmed that dietary supplementation of betaine reduced plasma homocysteine concentrations in blood and improved bone strength [37]. Moreover, adding other nutritional elements with betaine enhanced the laying parameters in heat-stressed birds. For instance, the improved performance of Fayoumi laying hens reared under high environmental temperature was reported after dietary supplementation with betaine, folic acid, and choline [82]. Laying hens aged 32–48 weeks under chronic heat-stress conditions showed significant enhancement in feed intake, productive traits, and laying performance after dietary addition of antioxidants, including vitamins E (150 mg/kg) and C (200 mg/kg) with betaine (1000 mg/kg) [83]. Dietary inoculation of betaine at a 400 mg/kg level could improve the hen-housed egg production, hen-housed laying rate, and layer chickens’ health under heat stress [84].

Heat stress could significantly reduce the concentrations of vitellogenin and very low-density lipoproteins in the egg yolk [85]; however, dietary betaine supplementation to 0.06% could increase these precursors and consequently egg production [86]. The serum levels of various hormones, including luteinizing hormone, estradiol, triiodothyronine, thyroxine, progesterone, and follicle-stimulating hormone, increased in laying hens when betaine was added to their diets [80,86,87]. Similarly, Zou and Feng demonstrated that dietary addition of 0.1% betaine increased the egg production of layer chickens by 10% as a result of promoting the production of follicle-stimulating hormone and luteinizing hormone and luteinizing hormones in the anterior pituitary gland [88]. Moreover, betaine increases the daily egg mass, serum estradiol, and melatonin concentration in the laying chickens.

##### Duck

Regarding the effects of betaine on ducks, it was found that supplementing their feed with betaine and their water with Vitamin C, as well as limiting their feed withdrawal (FW) can improve live weight gain (LWG) when exposed to heat stress. However, the LWG improvement was only 4.3–6.3% and did not affect feed intake [89]. As mentioned by another study, a diet supplemented with betaine at 700 ppm has beneficial effects on SCFAS, biochemical blood parameters, and body weight of broiler ducks under hot climate stress [77]. In another study, various feeding schedules of a diet supplemented with 800 ppm betaine have been investigated in 240 Cherry Valley (*Anas platyrhynchos*) meat ducks challenged by heat stress. The findings showed such a diet could enhance the growth performance of the ducks when fed twice daily [90]. In another study, supplying ducks’ drinking water with betaine at 400–1200 mg/L and exposure to heat stress have indicated positive modifications in the growth parameters, which could be associated with improved biomarkers of homeostasis in the blood [91]. In a recent investigation by Kumar et al., ducks’ exposure to heat stress and water supplemented with 300 mg/L Vitamin C and 800 mg/L betaine indicated an improvement of 52.7% in LWG and 18.5% in the feed-to-gain ratio [92]. In a study, ducks within the age range of 22–42 days were subjected to 34 °C under feed restriction (FR), with betaine supplementation (800 mg/L) and Vitamin C (300 mg/L). The results indicated improvement in LWG and feed intake compared to other treatments exposed to 34 °C with FR. The high temperature lowered LWG by 28.6%, and the combination of FR plus betaine limited the reduction to 10.2% [93]. Another study revealed that supplementing the diet of Domyati ducks with up to 1.5 g/kg of betaine could improve health, laying and hatchability parameters, semen quality, nutrient digestibility coefficients, and economic efficiency under summer conditions [94]. Comparing the effects of betaine (0.5 g/kg of diet) and dl-methionine (1.2 g/kg of diet) on carcass traits and growth performance has revealed that betaine is more effective than methionine at enhancing feed efficiency, growth, and carcass quality of starter ducks [64]. 

##### Quail

It was found that feeding quails a betaine-supplemented diet could increase egg weight by 3.82% while providing them with 2800 kcal/kg of metabolized energy and 17% protein [95]. A previous report has confirmed that supplementing a diet with betaine and 20% crude protein might also increase the weight gain of quails [96]. However, another study revealed that betaine supplementation had no effect on quails’ egg production at the beginning of the laying period [97]. Similarly, it was shown that supplementing a diet with betaine did not affect the egg production of quails during 42–63 days of age or on 50% of egg production after supplementing a diet containing 18.7% protein with betaine [98]. Ratriyanto et al., found that activated silicon dioxide plus betaine can be effectively added to the quails’ diets at the onset of rearing and laying periods [99].

##### Geese

Studies on betaine in poultry addressed chickens more than geese. The effects of dietary treatment with methionine and betaine on slaughter performance, serum biochemical parameters, and liver BHMT have indicated that optimal methionine dietary supplementation could elevate hepatic protein synthesis, growth performance, total protein, albumin, and globulin growth in growing goslings. Additionally, by upregulating the expression of the BHMT gene, betaine could replace methionine in the diet to enhance slaughter performance [100]. According to previous studies, betaine could significantly decrease the Landes goose abdomen lipid weight, improve the liver weight, and repair the hepatic injury, which can lead to positive economic impacts [101]. Regarding histological examination, geese treated with betaine had smaller lipid droplets than those on a high-carbohydrate diet. Groups treated with betaine indicated diffused lesions, increased microvesicular steatosis, swollen hepatocytes, and decreased macrovesicular steatosis. Dietary supplementation of betaine could distribute the fat content very well, but the weight of liver enhanced significantly [101]. Investigating the effect of betaine on changes in C/EBPβ gene transcription, lipid distribution, and DNA methylation in Landes geese’s liver indicated that betaine decreased the goose C/EBPβ gene expression, but did not play a direct role in regulating its methylation [102]. Dietary supplementation of betaine could reduce fat deposition and increase lipolysis in the finishing period of geese by reducing energy requirements and feed consumption [102].

#### 3.1.2. Turkey

There is a dearth of research on the effects of betaine on turkey. As indicated, 96% of infected birds older than 70 days responded favorably to treatment with 2.5 g of anhydrous betaine/liter of drinking water administered over two days [103]. The study by Dorra et al., revealed that supplementing the diet of grower turkeys with 150 g betaine per 100 kg/diet improved blood parameters, carcass features, growth performance, and protein metabolism, which was economically efficient [104]. However, the feed intake of turkey was not increased by supplementation of betaine in the diet. Adding 0.10% and 0.09% betaine to turkeys’ baseline diets had no effect on their performance. However, it could improve breast muscle yield and reduce body fat content [74].

#### 3.1.3. Carcass Traits

The addition of betaine at different concentrations to the diets could enhance breast muscle yield [79,105,106,107,108] and reduce the body fat contents of broiler chickens [109,110], turkeys [74], and ducks [64]. Carcass weights and dressing, thigh, breast, and giblets percentages were significantly improved by supplementing the diet with betaine at levels of 0.1 or 0.2% [110]. Similar improvements were reported in broiler chickens’ breast yield, total edible parts, and carcass after adding betaine to the diet [111]. Compared to a basal diet provided with choline plus methionine, when chicks received a basal diet supplemented with betaine plus methionine, their carcass fat levels were reduced [21]. Dietary inclusion of betaine for heat-stressed broiler chickens, especially in the summer season, increased breast muscle weight [112]. Likewise, ducklings who received diets supplemented with betaine (1.5 and 1.0 g/kg) displayed significant improvement in breast muscle yield [94].

However, dietary treatment with betaine did not significantly affect liver, gizzard, and giblets weights, but the abdominal fat decreased [32,62]. Moreover, adding betaine to broiler chickens’ diets induced no significant effect on carcass or breast yield and internal organs [45,113]. Carcass parameters, except the redness of breast muscle of broiler chickens, were not also affected by dietary betaine and/or creatine [57].

Betaine can be used to increase lean and decrease fat in broiler carcasses [21]. Betaine could be described as a carcass modifier due to its ability to reduce carcass fat content and increase muscle yield. The ability of betaine to increase lean formation and reduce fat deposition may depend on sex, age, and genetics. An increase in the muscle yield after supplementation with betaine may result from its methyl group donor activity, which is essential for the synthesis of methionine, cysteine, lecithin, and glycine for protein synthesis [14,21,105,114].

Betaine contributes to the reduction in fat accumulation in the carcass through different metabolic processes [115]. It affects lipid metabolism by provoking the oxidative catabolism of fatty acids through carnitine synthesis. Carnitine is essential for transferring fatty acids to the mitochondrial membrane to oxidize fatty acids [116]. Therefore, supplementation of betaine improves carnitine synthesis, which directly reduces carcass fat content. In addition, betaine can increase the production of phosphatidylcholine from phosphatidylethanolamine that affects liver fat metabolism, and accelerates fat removal from the liver [117]. The antioxidant role of betaine is played through scavenging reactive oxygen species free radicals. In broilers subjected to heat stress, the antioxidant properties of betaine reduced the tissue damage resulting from lipid peroxidation [19]. Therefore, oxidative stress and lipid peroxidation declined in broiler chickens who received betaine supplementation. When added to a diet for broiler chickens lacking in methionine, betaine (1 g/kg) can considerably enhance antioxidant defenses and meat quality while lowering lipid peroxidation in the chickens’ breast muscles [33]. Moreover, betaine reduces thiobarbituric acid concentrations in the muscles of broilers [33]. It was found that producing very-low-density lipoprotein might prevent fat deposition in the liver, stimulate its elimination from the liver [117], and regulate the hepatic cholesterol metabolism [118]. Increased lipase activity and decreased triacylglycerols concentration were observed in betaine-supplemented layer hens [80].

### 3.2. Osmo-Regulation

The osmotic-protective effect of betaine is associated with its dipolar zwitterion properties and its water solubility [119]. Betaine is a significant organic osmolyte that keeps the osmotic pressure in the intestinal epithelial cells in control [120]. Osmolytes are important in dehydration conditions since they reduce water loss against osmotic gradient [69].

Petronini et al., attributed the osmo-protective activity of betaine to its concentration in cell organelles, especially cells under osmotic and ionic stressors, thus replacing inorganic ions and protecting cell membranes and enzymes from inactivation by inorganic ions [121]. In addition, betaine has anti-apoptotic effects that promote the proliferation of cells in a hyperosmotic medium [122]. Betaine can reduce energy loss for ion pumping in cells under hyperosmotic pressure [123]. As betaine enhances intestinal cell proliferation, the enlarged cell’s epithelium could increase the nutrient absorption surface. Moreover, as a result of the protection of the intestinal epithelium by betaine, the feed efficiency and growth of broilers exposed to water salinity stress would be improved [59,71]. Some Gram-positive and Gram-negative bacteria can deposit betaine while being transferred from the environment [75]. Moreover, it can help maintain intestinal cell integrity, consequently protecting the beneficial microbiota from osmolarity stress [75].

Supplementation of drinking water with betaine can promote the retention of a high amount of water in birds exposed to heat stress [124] and, consequently, reduce the problem of excessive droppings and wet litter. Due to its osmoregulatory properties, betaine is vital in mitigating heat stress in poultry [28]. It was found that 2.5 g anhydrous betaine/liter of drinking water could successfully treat 96% of diseased birds aged over 70 days over 2 days [103]. Treatment of diarrhea in birds is critical to avoid alteration in osmotic balance and increase the litter moisture contents, consequently reducing the atmospheric ammonia level and decreasing the flock’s susceptibility to infection. Betaine could enhance the capacity of the intestinal cells to bind with water [125] and improve the tensile structure of chicks [126]. As betaine has zwitterionic properties, it acts as an osmolyte agent that helps the maintenance of cell water metabolism without affecting cell division.

### 3.3. Anticoccidiosis

Coccidiosis is an important enteric protozoan parasitic disease of poultry that causes adverse and severe economic losses [127]. The disease is associated with severe diarrhea and high mortality. Betaine can positively affect the water balance of broilers exposed to coccidiosis [128]. It has been found that betaine was more effective than methionine for broilers under coccidiosis infestation [108]. Betaine improves performance parameters in broiler chickens exposed to *Eimeria* (*E.*) *acervulina* infection [9]. Similar studies showed that betaine might enhance the coccidiostats effectiveness in terms of better performance of *E. tenella* and *E. acervulina* infested broiler chicks [129,130]. As reported, the anticoccidial activity of betaine to the direct partial inhibitory effect on the protozoon intestinal invasion and development or the indirect enhancement of the intestinal structure and function. Accordingly, betaine decreases the damaging effects of coccidiosis via inhibition of different developmental stages of *Eimeria* species and improves intestinal structure and functions. The reduction in villus height as a result of *Eimeria* infection was enhanced after the addition of 0.10% betaine to chicks’ diet [69].

Moreover, supplementing a diet with betaine decreased the crypt-to-villus ratio in coccidiosis-infested and healthy chicks, and it decreases the lesion score in coccidiosis-infested birds [108,125,126,131]. In a study by Augustine and Danforth, broiler chickens challenged with *E. acervuline* indicated digestibility enhancement of fat, methionine, protein, carotenoid, and lysine when fed betaine [132]. Nevertheless, betaine at a level of 1 g/kg of diet had no effect on *Eimeria* oocysts output or intestinal lesion score and *Clostridium perfringens* count in the ceca of broiler chickens [133].

### 3.4. Immune Modulation

Improved humoral immune response after supplementation with betaine in broiler chickens has been reported [69,125]. Dietary betaine significantly improved broiler chicks’ immune response to Newcastle disease virus (NDV) [134]. Farooqui et al., also demonstrated a high antibody response to NDV in broiler chicks supplemented with Vitamin C and betaine [135]. Similarly, dietary supplementation with betaine (1 g/kg) significantly improved the primary antibody titers against NDV and infectious bronchitis virus (IBV) in heat-stressed broilers at 27 and 35 days of age [22]. In addition, the humoral immune responses of *Mycoplasma gallisepticum* infected broiler chickens to living IBV, and infectious bursal disease virus vaccines were enhanced after water treatments with colistin and betaine, tylosin and betaine, and betaine alone [136]. In another study, the effect of betaine in water on the immune response of broiler chickens to inactivated NDV and avian influenza virus (AIV) vaccine was investigated. The obtained results indicated that hemagglutination-inhibition antibody values for both NDV and AIV vaccines were significantly higher in vaccinated and betaine-treated chickens than vaccinated non-treated chickens [137].

Moreover, betaine treatment could enhance the measured humoral antibody titers against sheep red blood cells under heat stress [138]. However, the dietary addition of betaine (0.121%) showed no significant enhancement of the humoral immune response in 42-day-old broiler chickens [139]. Moreover, it was found that in-ova inoculation of betaine and choline did not affect immunoglobulin (Ig) M, Ig G, and total antibody titers of hatched chicks [109].

It has been shown that betaine can stimulate the humoral immune response via regulation of cytokines production in the macrophages of the liver cell, inhibition of prostaglandin synthesis [140], as well as increasing the release of nitric oxide from heterophils and macrophages [69].

Exposure to heat stress leads to decreasing the number of lymphocytes (L), increasing the number of heterophils (H), and consequently increasing the H:L ratio [141]. This reduction in L count is related to the increase in inflammatory cytokines production and consequently stimulation of corticotrophin-releasing hormones in the hypothalamus [110]. Betaine decreases the H number and increases the L number. The study of Gudev et al. [142] revealed a lower H/L ratio in broiler chickens who received betaine at a concentration of 1 g/kg of diet, compared to the control group. In broiler chickens subjected to a high temperature, both hematocrit and H/L ratios were also found to be positively affected by betaine and/or creatine supplementation [57].

## 4. Heat-Stress Alleviation

Heat stress known as one of the effective challenges in industrial poultry farms. Given the absence of cutaneous sweat glands and feathers covering the body, avian species are very sensitive to heat stress. Panting is a typical respiratory evaporative heat-loss method in birds [143]. High temperature and humidity, especially in hot summer seasons, negatively affect the endocrine system, acid–base imbalance, organs’ functions, economic traits, and welfare [144]. The heat-stressed layer chickens usually show high mortalities [145] as well as decreased feed intake and performance [19,83,146], and immunosuppression of broiler and layer chickens [147].

However, several studies showed the heat-relieving effect of betaine in exposed broiler chickens [25,26,60]. Betaine efficiently improved thermo-physiological performance and alleviated chronic heat stress in broiler chickens [20,57,148]. Awad et al., stated that nutritional properties of betaine may help Domyati ducks to fight against poor management and heat-stress conditions [94]. Both betaine and Vitamin C proved to have similar effects in relieving the negative effect of high temperature on the growth traits of chickens [19,135]. Attia et al., reported that adding vitamins C, E, and betaine to the diet of laying hens alleviated the negative impacts of birds under chronic heat stress [83].

Heat-stressed birds recorded long tonic immobility due to a high-intensity level of fear [149]. Betaine supplementation can reduce the fear response of heat-stressed broiler chickens and consequently decrease tonic immobility duration [25]. Heat-stressed broiler chickens who received betaine at a concentration of 2 g/kg diet showed a shorter period of tonic immobility than control, indicating an improvement in chickens’ welfare [68]. The positive impact of betaine on tonic immobility can reduce stress on birds [137]. The positive effects of betaine on tonic immobility can minimize stress on broiler chickens [150]. Recent research indicates that supplementing a broiler chicken diet with betaine (0.1%) could decrease the negative effects on performance parameters and boost production capacity when exposed to environmental heat stress [151]. This means that betaine can serve as a profitable nutritional strategy for alleviating heat stress.

## 5. Conclusions

Supplementation of poultry diets with betaine has been advantageous in improving the productive performance of layer and broiler chickens as well as other species of birds, enhancing the carcass traits, the osmotic pressure, and the immune response, and alleviating the coccidial and the negative effects of heat stress on broiler flocks. Betaine, therefore, can be used to enhance the health and productivity of poultry for enhancement of physiological conditions and metabolism, as well as improving the immune response to vaccination against stressful viral infection. Further research work and studies are necessary to explain the different mechanisms and modes of action of betaine in birds’ bodies.

## Figures and Tables

**Figure 1 animals-12-02624-f001:**
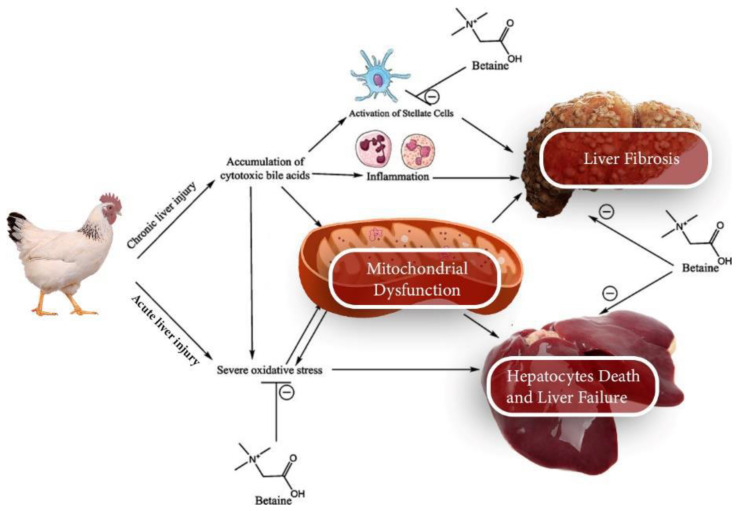
The mechanism of betaine in poultry liver injuries (the figure is designed by the authors of the current study).

## Data Availability

All prepared data are presented in the present article.

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
