# Peer review of "Betaine: A Potential Nutritional Metabolite in the Poultry Industry"

_animals, 2022, doi:10.3390/ani12192624_

Round 1

Reviewer 1 Report

The comments:   General Comments: 1. Betaine is a potential nutritional metabolite that has played a vital role in poultry nutrition and industry in recent years.  2. In the title "interaction of betaine with other elements": The author discussed methionine, choline, and creatine but the role of Betaine in the synthesis of carnitine is not discussed enough.  3. In the title "anticoccidial effect": The authors discussed on anticoccidial effects of betaine but the question is... if betaine has anticoccidial effects so it maybe has some effects on other parasites too which can be more searched and discussed.  4. line 279: in-ova inoculation of betaine should be  in ovo injection of betaine 5. In the title "Immune-modulatory": Betaine has good effects on ND vaccination, what about the effects on other vaccines? As we know vaccination is a stressful condition and if it causes a positive reaction to the stress conditions so it can be more helpful in the vaccination programs and the authors can discuss it in more detail. 6. In the title "Heat stress alleviation": The authors should indicate the breed of birds clearly when they discuss the effects of betaine on the heat stress of the birds/chickens. Seven times the authors mentioned that it has positive effects on different birds (which bird/breed?)  7. In the title "Conclusion": The authors repeat some sentences of text and abstract and it is better if they write this paragraph with different sentences. 8. In the title "Conclusion: The authors can mention the effects of betaine on vaccination stress briefly in this section too. The most important stress in health farms is vaccination and any component that can reduce this stress should be more attended.      Comments for editor: 1. The article has a good design and it is well written. 
2. The quality of language is acceptable.  2. I believe this review article has a high potential for receiving many citations in the next few years

Author Response

Dear Respected Reviewer, 

Thank you very much for expending the time for reviewing this article and mentioning some important comments.

I tried to revise the manuscript based on your comments and the revisions are highlighted in red color.

The answers to 8 comments.

  1. Thank you for your comment. Done
  2. Done as requested. Lines 96-110, 119-137.

  3. To our knowledge and after a comprehensive search, the effect of betaine supplementation on other parasite infestations has not been investigated.
  4. Done as requested. 

  5. Done as requested. 
  6. Done as requested. Lines 147, 153, 193, 196, 199, 220, 224, 342, 388.
  7. Done as requested.
  8. Done as requested. Lines 436-445.
  Please let me know if the article needs more revisions.  

Best regards

Dr. Daryoush Babazadeh

Reviewer 2 Report

In my opinion, a paragraph was missing about when it is advantageous to use betaine, as we know the advantages of methionine and its metabolites.

Author Response

Dear Respected Reviewer, 

Thank you very much for expending the time for reviewing this article and mentioning some important comments.

The manuscript has been strictly revised by considering all reviewers' comments carefully. All new revisions are written in red color.

Please let me know if the article needs more revisions.

The number of lines can be different based on the track revision requested by the editor of the journal after final revisions. 

Best regards

Dr. Daryoush Babazadeh

Reviewer 3 Report

Dear Authors,

Please see my comments below:

General comments:

1. Affiliation of author one is missing 

2. Most of references are outdated. Please update them to recent studies, perhaps years 2015 and after. There are many papers (free online access) out there worked on the effects of betaine.

3. The manuscript requires to be checked for grammatical errors. 

4. Add more keywords such Osmolyte etc. 

5. There are minor formatting issues. For example: number each section and sub-sections: 2. Performance and 2.1 chicken 2.2 duck etc

6. The manuscript needs to be checked for percentage of similarity. 

Major and minor comments:

1. Line 49. add reference.

2. Line 56. One of the example of using outdated references. " The efficacy of betaine decreases at higher concentrations [14]". The study was done in 1999 and there are many studies showed betaine at higher dosages has additional benefits. Revise the statement with recent studies. 

3. Line 63. add system to "The immune" 

4. Figure 1 needs citation. 

5. Line 82. "homocysteine to methionine". Homocysteine needs to be introduced and why converting it has beneficial impacts. 

6. Line 84. Abbreviations must be introduced for the first time. 

Line 89. "Dietary betaine and methionine in broiler chicks' diet can replace each other".  Please reword it, maybe: betaine can be used as an alternative to Meth?

7. Line 113. Limit your citations to 2 or maximum 3 if it requires.  Check the whole manuscript.

8. Line 124. Discuss other dosages of betaine (0.5g-2.5g). 

9. Line 132. Break down the references, and move each of them to where it belongs. 

10. Line 135. It mentioned several time in the text that Betaine improves growth performance. Avoid repetitive statements. 

11. Line 151-153. it mentioned betaine has no benefits while it discussed earlier it has benefits. I would recommend you to re-organize your statements and move them in one place. 

12. Line 181. why Vitamin C discussed here? The review is all about betaine. 

13. Line 201. 1.5 g B/kg?

14. Lines 203-207.what dosages of betaine they used for the study?

15. Line 224. Introduce the abbreviations. 

16. Lines 228-229. Be consistent. Why having P-values for this section only. 

17.  Line 388. Vitamin C?

18. Line 403. Change it to: the negative effects of  heat stress on broiler.

Regards, 

Author Response

Dear Respected Reviewer, 

Thank you for the time that you spend on evaluation and recommendation. The manuscript has been strictly revised by considering all reviewers' comments carefully.

The number of lines may be different regarding the new revisions and the request of the journal editor for changing the format of the revised manuscript before submission.

Please let me know if the article needs more revisions.

Best regards

Dr. Daryoush Babazadeh

Round 2

Reviewer 3 Report

Dear Authors, 

Thanks for the revised version. The manuscript has been improved accordingly. However, there are still a few minor issues, listed below: 

1. No keywords have been added to the version I received. But, the authors mentioned they have added "Added (Immunity, Heat stress, Osmolyte)"

2. Figure 1 needs citation underneath. If the figure is from another study, it needs to be mentioned. 

Regards, 

Author Response

Dear Respected Reviewer,

Thank you very much for your constructive comments. All issues are well-addressed.

Best regards
Daryoush Babazadeh
